# Adenocarcinomas of the Lung and Neurotrophin System: A Review

**DOI:** 10.3390/biomedicines10102531

**Published:** 2022-10-10

**Authors:** Alberto Ricci, Claudia Salvucci, Silvia Castelli, Antonella Carraturo, Claudia de Vitis, Michela D’Ascanio

**Affiliations:** UOC Respiratory Disease, Sant’Andrea Hospital, Sapienza University of Rome, 00189 Rome, Italy

**Keywords:** adenocarcinoma, lung, neurotrophins, tyrosine kinase receptors

## Abstract

Neurotrophins (NTs) represent a group of growth factors with pleiotropic activities at the central nervous system level. The prototype of these molecules is represented by the nerve growth factor (NGF), but other factors with similar functions have been identified, including the brain derived-growth factor (BDNF), the neurotrophin 3 (NT-3), and NT-4/5. These growth factors act by binding specific low (p75) and high-affinity tyrosine kinase (TrkA, TrkB, and TrkC) receptors. More recently, these growth factors have shown effects outside the nervous system in different organs, particularly in the lungs. These molecules are involved in the natural development of the lungs, and their homeostasis. However, they are also important in different pathological conditions, including lung cancer. The involvement of neurotrophins in lung cancer has been detailed most for non-small cell lung cancer (NSCLC), in particular adenocarcinoma. This review aimed to extensively analyze the current knowledge of NTs and lung cancer and clarify novel molecular mechanisms for diagnostic and therapeutic purposes. Several clinical trials on humans are ongoing using NT receptor antagonists in different cancer cell types for further therapeutic applications. The pharmacological intervention against NT signaling may be essential to directly counteract cancer cell biology, and also indirectly modulate it in an inhibitory way by affecting neurogenesis and/or angiogenesis with potential impacts on tumor growth and progression.

## 1. Introduction

Lung cancer is the leading cause of death from cancer worldwide. There are two main histopathological forms: non-small cell lung cancer (NSCLC) (adenocarcinoma-ADK-squamous cell carcinoma and large cell carcinoma) and small cell lung cancer (SCLC) displaying neuro-endocrine features. The former represents about 80% of diagnosed cases. In the past few years, the incidence of NSCLC continued to rise. The average survival after five years is about 16%. Often, the delay in diagnosis precludes the possibility of a surgical approach [1], and chemotherapy is the only possible therapeutic strategy in the most advanced stages. The ineffectiveness of traditional chemotherapy treatments highlights the need to identify molecules that guide the growth, survival, and resistance to the treatment of neoplastic cells. Recently, progress has occured in this area of research with the identification of target molecules with the development of target therapies.

ADK of the lung represents the most common type of lung cancer and comprises about 40% of all lung cancers, regardless of smoking history or gender. It originates from small airway epithelial, glandular, and/or type II alveolar cells. Advances in lung ADK research have led to the dramatic modification of classical cancer treatment from standard cytotoxic chemotherapy and/or radiotherapy to mutation-target therapies. Target therapies often produce rapid tumor regression with few side effects, limiting the potential therapeutic efficacy to sensitive cancer cells, since normal cells lack the tumorigenic mutation. Identification of novel target mutations or molecules that can potentially control cell growth and that could be pharmacologically modulated is essential to overcome the development of drug resistance and to have different therapeutic weapons to be used in subsequent lines of treatment or as combination therapies.

Since the discovery of tyrosine kinase receptors (TRKs) in the 1960s, and their classification into different families, their relevant role in numerous aspects of cell biology has been clarify. At the same time, it became evident how their dysfunction could play a decisive role in different cellular biological activities, as well as the development of tumors, conditioning their biological behavior from the early and throughout the advanced stages of the disease. The discovery of TRKs mutations has allowed the development of new drugs able to inhibit these receptors and their downstream pathways. It represents a therapeutic innovation, a milestone in cancer therapy, able to impact mortality. Therefore, many TRK receptor inhibitors are widely used in the treatment of different human malignancies. Furthermore, molecular research is looking for new mutations and drugs for cancer treatment that can improve the patient quality of life and survival.

The development of neoplastic tissue is typically affected by local conditions that guide its development and progression. Among these, metastasization is certainly the most important event. Consequently, the interaction between neoplastic tissue and the microenvironment is fundamental to favor these events [2]. In the 1970s, the concept of neo-angiogenesis was defined as the ability of neoplastic tissue to produce new vessels necessary to cover its needs [3]. As a result, the research sought to identify the biological factors underlying this process and to counteract them [4,5].

Another important endothelial homeostatic mechanism essential for cell growth is represented by neo-lymph angiogenesis [6]. Furthermore, the role of several lymphangiogenic factors influencing metastatic disease has been demonstrated [7].

Impaired oxygen demand during solid tumor proliferating cells generates a hypoxic intra-tumoral environment. This condition activated hypoxia-inducible factor (HIF) that upregulates the expression of several pro-angiogenetic factors [8,9], therefore, HIF has a critical role in neo-angiogenesis. Furthermore, hypoxia and BDNF induce HIF expression, while HIF is a transcriptional activator of the TrkB NT receptor gene [10]. The expression of the low-affinity NT receptor p75 increased the stabilization of HIF resulting in an increase in migration, invasion, and stemness in response to hypoxia in some cancer cell types [11].

In the same way as the neo-formation of blood and lymphatic vessels, there is the possibility, for the correct tissue homeostasis, to promote the development of nerve endings within the tumor. This condition has been defined neo neurogenesis [12].

Simplistically, the role of peri- and intra-tumoral neuronal innervation had been exclusively considered mechanical. More recently, tumor innervation has been shown to have a functionally relevant role. It appears to be essential in regulating the complex interstitial network, the local immunological response, neo-vascularization, and the behavior of tumor cells [13,14,15,16,17].

Nerve fibers development within the tumor mass could also serve to release mediators able to direct tumor cell behavior [15]. This condition favors beneficial mutual assistance between the nervous system and neoplastic cells mediated by the release of mediators both from the nerve terminals and the tumor cells, with cancer neurotrophic and axon-guidance activity. Perineural invasion, axonogenesis, chemoattractant, and survival molecules secretion support this relationship. Furthermore, the presence of neuronal markers has often been considered a prognostic factor in several human malignancies [18,19,20].

Several lines of evidence support the hypothesis that cancer cell support neurogenesis by secreting different neurotrophic factors such as nerve growth factor (NGF), brain-derived neurotrophic factor (BDNF), and neurotrophin 3 (NT3). These mediators and the expression of their receptors are sometimes associated with cancer prognosis [21].

This review aims to provide a comprehensive picture of the literature data and a deeper insight into the possible role of neurotrophins (NTs) and NT receptors within lung adenocarcinoma cell biology. Moreover, we provide a clear view of the available data to highlight novel molecular mechanisms useful for diagnostic but, above all, therapeutic purposes and how they may be considered novel therapeutic targets. This is because ADK, which represents the leading cause of cancer death worldwide, displays several different gene mutations actually under investigation.

To identify relevant published data on NTs and NT receptors in lung adenocarcinoma, we conducted a literature search on PubMed using the following keywords: neurotrophins, NT receptors, nerve growth factor, Brain-Derived Neurotrophic Factor, NT-3, lung adenocarcinoma, and human”. In Table 1, we reported the search strategy conducted.

Inclusion criteria used to select manuscripts derived from the PICO (problem, intervention, comparison, and outcome) and PIPOH (population, intervention(s), professionals, outcomes, and health care setting/context) structures. Therefore, the studies used are all experimental and published in the last 20 years. Exclusion criteria are the studies published in a language other than English and the use of a qualitative methodology. About 190 manuscripts were selected and evaluated for this review. The selected papers are listed in the reference section.

## 2. Neurotrophins and Neurotrophin Receptors in the Lung

NTs are a super-family of trophic growth factors that hold essential functions for the development, survival, and health of the central and peripheral nervous systems. Moreover, NTs have an important role outside the nervous system. More than 60 years ago, the prototype of NTs, the nerve growth factor (NGF) was discovered, and its function was identified. The discovery of NGF laid the groundwork for the research of novel NTs. Today, the mammalian NT family comprises NGF, brain-derived neurotrophic factor (BDNF), NT-3, and 4/5 [22]. NT-4 was only a specie variation of the NT-5, characterized in vertebrates, and the terminology NT-4/5 was usually applied. Conversely, NTs 6 and 7 were identified in fish but not mammals [23]. They are closely related molecules with about 50% of sequence identity with 90% homology among humans, mice and rats [24,25]. All NTs are synthesized as precursors with an N-terminal signal peptide and are further cleaved into mature NTs [19,26]. Their activities are mediated by the binding to two types of NT receptors: the low-affinity (p75) and the high-affinity tyrosine kinase NT receptors. The ability of these factors and their receptors to control cell growth and differentiation allows us to indicate a connection to cancer cell growth from a scientific and therapeutic point of view.

NTs and NT receptors are widely present in the normal lung. NTs produced by target cells (epithelium, smooth muscle cells, immune cells, etc.) not only regulate and address nerve-ending development and homeostasis, but they may sustain cell behavior, survival, development, or death in different physiological or pathophysiological conditions. Although the epithelium of the lung constitutively expresses NGF, BDNF, and NT3 [27], less information has been reported on NT receptor expression at this level. NGF, via the high-affinity TrkA receptor, seems to possess a survival effect on airway epithelial cells [28]. NGF is an essential survival factor for bronchial epithelial cells during respiratory syncytial virus infection. On the other hand, BDNF and NT3 through TrkB and TrkC receptors induce nitric oxide production via elevation of both intracellular Ca^2+^ concentration and endothelial nitric oxide synthase phosphorylation [29].

## 3. Neurotrophins and Their Receptors in Cancer

NTs have been investigated in different human malignancies in which they are frequently associated with cancer growth and progression. Overexpression of NTs and their receptors was demonstrated to promote epithelial to mesenchymal transition (EMT), symmetric division, self-renewal and plasticity of cancer stem cells (CSC). CSC are a small population of cancer cells that possess the capability of differentiation, tumorigenicity, ability to generate metastasis, and resistance to treatment. Their clinical relevance has been addressed. Therefore, targeting CSC may be a critical way to better understand the mechanisms of tumorigenesis, counteract cancer growth and develop novel anti-cancer strategies to improve patients’ outcomes. In this contest, NTs have demonstrated a significant role in promoting CSC clonal survival. Additionally, aberrant NT/NT receptor signaling has been recognized as a driver of cancer progression [30,31].

Many studies have underlined the possible role of NTs and their receptors as potential factors able to drive neoplastic cell growth, progression, and metastasis in different malignancies. In addition, their potential value as diagnostic and prognostic biomarkers have been debated. In tumors of the gastrointestinal tract, elevated levels of BDNF and its TrkB receptor are correlated with more severe disease and poor prognosis [32,33]. Contrarily, low BDNF plasma levels have been detected in colorectal cancer patients compared with healthy subjects suggesting a role of this factor in this malignancy as a biomarker [34] or as a marker of improved survival [35]. In prostate cancer, proNGF expression was correlated with the degree of malignancy (expressed as the Gleason score) and the ability to promote neurogenesis in the tumor microenvironment [36]. In ovarian cancer the expression of NGF, TrkA, and p75 receptors is also increased, suggesting clinical implications [37]. Similarly, the expression of proNGF is increased in thyroid tumors compared to normal healthy thyroid tissues, this makes its assay useful as a possible biomarker of malignancy [38]. Furthermore, in thyroid cancer the increased expression of NT high-affinity TrkA receptor has been correlated with tumor progression and lymph node invasion. p75 NT receptor expression is also increased in this malignancy [39]. More recently, the increased expression of different NTs and their receptors have been demonstrated in different lung cancer histological subtypes [40]. In addition, Trk fusion proteins are also detected in NSCLC, emphasizing the potential of NT and the NT receptor as targets for further therapeutical applications [41].

The NTRK gene family contains three members, NTRK1, NTRK2, and NTRK3, which produce TRKA, TRKB, and TRKC proteins. Intrachromosomal or interchromosomal rearrangement of these genes by the fusion of two genes together, may generate aberrant Trk proteins which in turn may lead to uncontrolled cell growth. The Trk gene fusion generates fusion proteins with intact functional tyrosine kinase domain, which leads to a persistent upregulation of downstream signal pathways. Trk gene fusion may be associated with oncogenesis in different cancer cell types. Recently, inhibitors with potential tolerability and efficacy are under investigation in specific clinical trials, in multiple cancer cell types [42,43,44].

These fusions can be detected in a broad range of human solid tumors, and in about 5% of NSCLC by next-generation sequencing, immunohistochemistry, DNA fluorescence in situ hybridization (FISH), and polymerase chain reaction (PCR). The primers used in RT-PCR for each neurotrophin and receptor are listed in Table 2, while Table 3 lists the antibodies to detect them via immunohistochemistry.

Despite the low percentage of Trk gene fusion among lung cancer patients, the high prevalence of this disease makes this option relevant, considering these oncogenes as possible targets in lung cancer [45].

Recently in lung cancer, two different gene fusions involving the NTRK1 gene were observed with the synthesis of constitutive TrkA tyrosine kinase domain activation [44]. A small percentage of patients affected with NSCLC harbored this mutation. Furthermore, 23 newly NTRK1, NTRK2, and NTRK3 gene fusions were identified across different tumor types, including lung adenocarcinomas. Thus, NTRK fusion represents a low-frequency but pan-cancer event that may drive a significant number of patients who may have benefited from anti-NTRK-inhibitors. Furthermore, the development of NTRK mutations using NTRK inhibitors has been reported [44,46].

Trk receptor activation resulting from the autophosphorylation of tyrosine residues in their intracellular tails, triggers several downstream signaling pathways involving enzymes and adaptors such as extracellularly regulated kinases (ERK), the phosphatidyl inositol kinase 3 (PI3K), and phospholipase C (PLC). All these pathways regulate cell proliferation, differentiation, and survival in neuronal and non-neuronal cells. Differently, signaling via p75 receptors leads to a c-jun N-terminal kinase (JNK) cascade depending on the specific adaptor proteins bound to the receptor. Activation of p75 results, via JNK, in p53 activation and expression of pro-apoptotic genes (Bcl-2). Furthermore, the p75 receptor, in combination with tumor necrosis factor TRAF6, promotes activation of nuclear factor-kB (NF-kB) signaling with a pro-survival effect. The variability of activation signaling promoted by NT receptors is a function of complexity that NT receptor exert outside and within the central nervous system, as well as in different pathophysiological conditions.

### Low Affinity NT p75 Receptor in ADK of the Lung

The low affinity p75 receptor was the first member of the tumor necrosis factor (TNF) receptor superfamily to be discovered. p75 is a glycosylated transmembrane receptor whose activation elicits several biological functions by interacting with its cognate ligands. p75 is an unusual member of TNF receptors for its ability to dimerize and its ability to act as a tyrosine Kinase (TRK) co-receptor. Isoforms of p75 receptor have been identified with truncated forms produced by alternative splicing and proteolysis. An example is a p75 receptor variant lacking exon III generating a receptor unable to bind NTs. p75 receptor may be activated by a constitutive metalloproteinase, resulting in a soluble extracellular domain able to bind neurotrophins, and a receptor fragment containing the intracellular domains. Its specific function is uncertain, however, the soluble form of p75 is produced at high levels during development and nerve injury [47,48,49].

Complex p75 crosstalk with TRK receptors has been documented. It attenuates TRKs activation by non-preferred ligands. The relationship between p75 and TRKs is not one-sided [50].

p75 receptor induces apoptosis in the developing nervous system in the synaptogenic period. Furthermore, p75 can mediate a variety of biological functions that include cell survival, migration, cell invasion and proliferation, as well as cell death [51].

Therefore, a precise p75 activity is contradictory. Its specific function, under physiological condition, is uncertain and the role of the receptor activation still appears enigmatic. However, its activation could carry out, in neoplastic cells, interesting and important effects both from a scientific and a therapeutic point of view. p75 signal transduction may be influenced by mutual interaction between p75 and TRK receptors, the appropriate concentration of neurotrophic factors at the binding site, the different cytoplasmic receptor expression. Consequently, it is not possible to generalize the role of p75 in any biological condition and thus, in cancer biology. Moreover, the p75 receptor is often absent in many malignancies and its expression is not considered an indication of cancer phenotype, although it can be used as a marker in some clinical conditions. More recently, some authors indicate cancer cells possessing p75 expression display some cancer stem cell characteristics such as self-renewal and resistance to chemotherapy [26,48].

The data suggests, in gastric carcinomas, that the expression of p75 can inhibit metastasis by down-regulating the expression of MMP9 and up-regulating TIMP1 through NF-kB signal transduction pathway is not in line with these observations [52,53].

## 4. Neurotrophins and Their Cognate High Affinity TRK Receptors in ADK of the Lungs

NTRK 1–3 genes encode the high affinity 140 kDa Tyrosine Kinase (TRK) receptors. Different NTs preferentially bind to specific high-affinity TRK receptors: NGF preferentially binds to the TRKA receptor; BDNF and NT-4 to TRKB; and NT-3 to the TRKC receptor. Although the binding to cognate receptors is considered to occur with a high affinity, it is regulated by receptor dimerization, receptor structural changes, or association with the low-affinity p75 receptor that is considered a co-receptor, enhancing the affinity and specificity modulating the activity of the TRK receptors. Furthermore, the ratio in NT receptor expression is essential in addressing the effect of NTs on cell activity and destiny. Competitive or synergic activity may be displayed [40,54].

### 4.1. NGF/TrkA Receptor Signalling

TrkA expression has been considered a marker of favorable prognosis in neuroblastomas [55]. On the contrary, in breast cancer, NGF/TrkA signaling contributes to cancer progression via the activation of ERK, SRC, and AKT pathways similar to gastric and pancreatic cancer [56].

Although the anatomical distribution within cancer cells is still unclear, the expression of NGF and TrkA are high in NSCLC [40,56]. In lung adenocarcinomas, the expression and role of the NGF/TrkA system has been demonstrated [19]. The effect is mediated by NGF binding to TrkA receptor via activation of the anti-apoptotic protein Akt, and it is blocked by NT receptor inhibitor K252a. Furthermore, in a series of histologically different non-small cell lung cancers, the expression of TrkA and NGF was mainly documented in squamous cell carcinomas and to a lesser extent in adenocarcinomas [56]. Meanwhile, the overexpression of NGF and the increased release of hypoxia-inducible factor-1alpha (HIF-1a) magnify tumor neo-angiogenesis, and this correlated with microvascular density [5].

### 4.2. BDNF/TrkB Receptor Signalling

BDNF is a growth factor usually related to existing neuron survival, facilitating regeneration and synapse plasticity. Its increased expression has often been described in different cancer types. Squamous cell carcinoma and adenocarcinoma of the lung express more elevated levels of BDNF at both protein and mRNA levels. Its activity is mediated by the TrkB receptor and activates several downstream signaling such as PI3K/AKT, RAS/ERK, Jak/STATPLC/PKC, and AMPK/ACC [57]. TrkB is essential to BDNF function because TrkB deficiency in an ADK model reduces cancer’s ability to generate metastasis [58]. Cooperation between TrkB and EGFR signaling has been reported that may favor cancer cell dispersion and migration [59].

BDNF was produced and secreted by a percentage of lung tumors, with autocrine and/or paracrine effects [58].

The role of the TrkB receptor has been characterized in neural tumors, such as neuroblastomas, where TrkB expression represents a sign of aggressive behavior. The TrkB (also defined as NTRK2) receptor is fundamental for neuronal development but is an independent prognostic factor in different malignancies. High expression of TrkB correlates with poor prognosis in non-neuronal tumors such as ovarian, pancreatic, prostate, and gastroenteric tract cancers. Furthermore, TrkB overexpression has been documented in metastatic cells, while TrkB detection in adenocarcinomas of the lung has been associated with poor prognosis [20,57,60].

TrkB activation by its ligand, the BDNF, promote resistance to chemotherapy by PI3K/AKT pathway. Moreover, in in vitro lung adenocarcinoma cell culture, BDNF stimulates the pro-survival pathway via AKT suppressed by TrkB receptor inhibitor [19].

The TRKB receptor can be found expressed in different isoform. The active form of the receptor requires the full-length status (TrkB-FL). The truncated TrkB isoforms T1 and T2 do not have a clear biological function but, probably, under specific conditions, they can act as dominant negatives [61].

However, several studies reported a role of the T1 isoform in tumorigenesis [61]. Its overexpression induces liver metastasis via activation of RhoA signaling [62]. Moreover, expression of the full-length TrkB receptor in lung ADK cells seems to be associated with the increased risk of developing brain metastasis [63].

The role of BDNF/TrkB signaling has been assessed in cancer stem cell cultures derived from ADK cells of the lungs. TrkB was highly-expressed in these cells, where it was able to affect spheroid morphology and efficiency. The TrkB inhibition via pharmacological or siRNA against TrkB causes loss of transcription factors linked to epithelial to mesenchymal transition (EMT), suggesting that TrkB is involved in the full acquisition of EMT in cancer cells [60].

In the opposite direction, the results show a reduction of TrkB expression in adenocarcinomas, with a better prognosis. These data suggest a role as a suppressor gene for TrkB. This effect could be associated with DNA hypermethylation in lung adenocarcinoma [64].

### 4.3. NT3/TRKC

NT3 is related to the maintenance of the central nervous system in adults and the development of neurons in the embryo. Knockdown of NT3 in mice results in a severe deficiency in limb movement [65].

NT3 behaves differently depending on the presence or absence of TrkC (NT3’s main receptor). For example, in neuroblastomas, overexpression of NT3 alone is related to a better prognosis, while the overexpression of both NT3 and TrkC leads to a poorer outcome. This is based on the hypothesis that TrkC acts like a proto-oncogene only when it is in the presence of a ligand (in fact, in the absence of a ligand, TrkC promotes cellular apoptosis) [66]. This highlights how fundamental the NT3/TrkC complex is in cancer development.

TrkC overexpression has been reported in patients with different cancer cell types from mesenchymal and epithelial cell lineage. Its aberrant activation and presence in NTRK3 fusion proteins induces the epithelial to mesenchymal transition program, increases the cancer cell growth rate and oncogenic capacity activating different pathways associated with tumor development and progression [64,67].

A consistent number of mutations in the kinase domain of the TrkC receptor have been demonstrated that imply the possible constitutive activation of the TrkC receptor with acquired resistance to Trk-inhibitors [68].

## 5. Therapy in NTRK Positive Solid Tumors

Nowadays, only two drugs are approved for NTRK+ lung cancer: Entrectinib and Larotrectinib.

Entrectinib is approved for solid tumors with NTRK gene fusion and NSCLC with ROS1 mutation (first line of therapy), administered at the dosage of 600 mg daily in adults and 300 mg/m^2^ daily in children > 12 years old [69]. It is generally well tolerated, however the most common adverse effects (in most cases grade 1–2) are dysgeusia, fatigue, constipation, diarrhea, edema peripheral, dizziness, nausea or vomiting, paresthesia, increasing levels of creatinine or transaminases, myalgia, anemia, and increased body weight [70]. Cases of developing drug resistance have also been described [71].

Larotrectinib has been approved only for solid tumors with NTRK gene fusion, the dosage for an adult is 100 mg twice/day and in children >12 years old it is 100 mg/m^2^ twice/day. It can cross the blood-brain barrier, so it is effective against central nervous system metastasis [72]. Adverse effects are similar to those of Entrectinib, with the addition of a possible reduction in neutrophil count [73].

Other drugs under investigation for Lung Cancer with NTRK mutations are Selitrectinib (ClinicalTrials.gov Identifier: NCT04275960), Repotrectinib (ClinicalTrials.gov Identifier: NCT04094610), Taletrectinib (ClinicalTrials.gov Identifier: NCT02675491), Belizatinib (ClinicalTrials.gov Identifier: NCT02048488), Altiratinib (ClinicalTrials.gov Identifier: NCT02228811). In Table 4, we reported TRK inhibitors clinical trials in NTRK fusion-positive solid tumors, including NSCLC.

## 6. Conclusions and Perspectives

Great effort in lung cancer research has focused on transformed cancer cells. This led to the identification of important pathways and specific genes involved in oncogenesis, such as EGFR, KRAS, AKT, and ROS1 [74]. However, less attention has been given to the role of the tumor microenvironment and the factors able to influence it, orienting tumor behavior, the ability to generate metastases and resistance to therapy. NTs could directly interact with tumor cells, expressing NT receptors. Indirect interactions of NTs and tumor cells could also occur, resulting from the interplay between malignant cells and tumor microenvironment, orienting phenomena such as neo-angiogenesis, neo lymph angiogenesis or neo neurogenesis, essential for tumor growth. In this context, the relationship between NTs and HIF-1a or VEGF-c the key regulators of tumor lymphangiogenesis and metastasis, are known [22,75]. In addition, NTs may modulate the immune system involved in the fight against neoplastic development. Data from the literature support the hypothesis that NTs signaling counteracts anti-tumor immunity and immunotherapy response [76]. Moreover, NGF participates in tumor immune surveillance, modulating both innate and adaptative immune responses [77,78]. These observations, yielding new perspectives and a basis for understanding some mechanisms that regulate tumor cell behavior, suggest the possibility of considering these growth factors as potential therapeutic target molecules. From this point of view, in different adult and pediatric tumors, NTRK gene fusions are identified as oncogenic drivers [79,80,81]. In NSCLC, the prevalence is below 5%. In Table 5 and Table 6 we showed different NTRK gene fusions and their reported frequency of in NSCLC. Trk fusion is not linked to specific clinical features but is mutually exclusive with other mutations. Trk inhibitors, such as larotrectinib and entrectinib, display efficacy and a safe profile and they are approved for the treatment of NT Trk fusion-positive tumors. Furthermore, next-generation NT Trk inhibitors, selitrectinib, repotrectinib, and taletrectinib, are generated to overcome acquired resistance. Much is still unknown about the complex NT mechanisms that control cancer cell survival and cell growth. The current idea is that NTs participate, in parallel with other different factors, in lung cancer cell control. Cancer is a pathologic process that results from a variety of causes and molecular events. Within the next few years, further research and understanding will direct the assessment and treatment of lung cancer. ADK have had the biggest expansion in this regard. Although the percentage of ADK that express NTs and NT high and low-affinity receptors are rare and the prevalence of NT gene fusion account for approximately 5% of cases, collectively they affect a considerable number of patients. Analysis of this system could be routinely considered for possible target therapies. Considering the difficulty in treating advanced lung ADK that is not surgically resectable, and burdened by a high mortality rate, alternative therapies need to be established. Cancer can be treated by ongoing extended therapies that can control the disease for months or years. The emerging new paradigm is that small molecules and biologics will become important tools to favor the transition to chronicity in advanced cancer. This could happen by better understanding the pathophysiological mechanisms of the disease, and the possibility of combination therapies based on molecular targets. In this context, it is important to consider the NTs system as avenue for treating lung cancer.

## Figures and Tables

**Table 1 biomedicines-10-02531-t001:** Search strategy conducted.

((genomic OR gene) AND (profiling OR fusion OR rearrangement)) AND Lung AND cancer NOT (review[ptyp])
NGF or BDNF or NT-3 AND Lung AND cancer
TrkA or TrkB or TrkC or p75 AND Lung AND cancer
((genomic OR gene) AND (profiling OR fusion OR rearrangement)) AND (“lung cancer” OR “lung adenocarcinoma”) NOT (review[ptyp])
((genomic OR gene) AND (profiling OR fusion OR rearrangement)) AND cancer NOT (review[ptyp])
(TRK or NTRK or NTRK1 or NTRK2 or NTRK3 or tropomyosin) AND (fusion or rearrangement) NOT (review[ptyp]))

**Table 2 biomedicines-10-02531-t002:** Primers used in RT-PCR [19].

RT-PCR	Primers
NGF	5′CGCTCATCC-ATCCCATCCCATCTTC, 3′CTTGACAAGGTGTGAGTCGTGGT
BDNF	5′AGGGTTCCGGCGCCACTCCTGACCCT, 3′CTTCAGTTGGCCTTTGTGATACCAGG
NT-3	5′CGAAACGCGTATCGCAGGAGCATAAG, 3′GTTTTTGACTCGGCCTGGCTTCTCTT
TrkA	5′TCTTCACTGAGTTCCTGGAG, 3′TTCTCCACCGGGTCTCCAGA
TrkB-FL	5′TCTTCACTGAGTTCCTGGAG, 3′TTCTCCACCGGGTCTCCAGA
TrkB. [TR-]	5′TAAAACCGGTCGGGAACATC, 3′ACCCATCCAGTGGGATCTTA
TrkC	5′CATCCATGTGGAATACTACC, 3′TGGGTCACAGTGATAGGAGG
p75	5′AGCCCAC-CAGACCGTGTGTG, 3′TTGCAGCTGTTCCACCTCTT

**Table 3 biomedicines-10-02531-t003:** Antibodies often used for immunohistochemistry assays [19].

Immunohistochemistry	Antibodies	Company
NGF	rabbit anti-NGF polyclonal antibody	c-548; Santa Cruz Biotechnology, Santa Cruz, CA, USA
BDNF	rabbit polyclonal antibody anti-BDNF	sc-546; Santa Cruz Biotechnology, Santa Cruz, CA, USA
NT-3	rabbit polyclonal antibody anti NT-3	c-547; SantaCruz Biotechnology, Santa Cruz, CA, USA
TrkA	rabbit polyclonal TrkA immunoglobulin	sc-118; Santa Cruz Biotechnology, Santa Cruz, CA, USA
TrkB-FL	rabbit polyclonal TrkB immunoglobulin	sc-012; Santa Cruz Biotechnology, Santa Cruz, CA, USA
TrkB. [TR-]	rabbit polyclonal TrkB [TK-] immunoglobulin	sc-119; Santa Cruz Biotechnology, Santa Cruz, CA, USA
TrkC	rabbit polyclonal TrkC immunoglobulin	c-117; Santa Cruz, Biotechnology, Santa Cruz, CA, USA
p75	goat polyclonal antibody to human p75 NT receptor	sc-6188; Santa Cruz Biotechnology, Santa Cruz, CA, USA

**Table 4 biomedicines-10-02531-t004:** TRK inhibitors clinical trials in NTRK fusion-positive solid tumors, including non-small cell lung cancer (NSCLC).

Study Name	Phase	Inhibitors	Population with NTRK-Fusion Positive Solid Tumors	Enrollment (n)
NAVIGATE (NCT02576431)	II	Larotrectinib	Adults and children	320 patients
STARTRK-2 (NCT02568267)	II	Entrectinib	*NTRK*-, *ROS1*- and *ALK*-fusion positive	300 patients
NCT01639508	I	Cabozantinib	*NTRK* fusion, or *MET* or *AXL* overexpression, amplification, or mutation	68 patients
NCT03215511	I/II	Selitrectinib	Adult and pediatric	93 patients
TRIDENT-1 (NCT03093116)	I/II	Repotrectinib	*NTRK*-, *ROS1*- and *ALK*-fusion positive	450 patients
NCT02675491	I	DS-6051b	*NTRK*- or *ROS1*-fusion positive s	15 patients
NCT01804530	I	PLX7486	*NTRK*-fusion positive	59 patients-discontinued
NCT02920996	II	Merestinib	*NTRK*-fusion positive or *MET*-mutation NSCLC	25 patients
NCT03556228	I	VMD-928	*NTRK1* alterations, including fusions, positive	54 patients
NCT02219711	I	Sitravatinib	*NTRK*-fusion positive NSCLC	260 patients

**Table 5 biomedicines-10-02531-t005:** Reported frequency of NTRK gene fusions in NSCLC.

Study	Histopathology	Frequency	NTRK	Fusion Partner(s)	Detection
Farago, 2018 [53]	NSCLC	0.23%	*NTRK1, NTRK3*	*SQSTM1,TPR, IRF2BP2, TM3, MPRIP, ETV6*	DNA NGS, RNA NGS or FISH
Vaishnavi, 2013 [50]	ADK without oncogenic drivers	3.3%	*NTRK1*	*MPRIP, CD74*	DNA NGS or FISH
Stranniki, 2014 [54]	ADK	0.19%	*NTRK2*	*TRIM24*	RNA sequencing
Miyamoto, 2019 [36]	Non-SqCC NSCLC	0.05%	*NTRK3*	Not reported	RT-PCR and NGS
Gatalica, 2018 [55]	ADK	0.1%	*NTRK1-3*	*TPM3, SQSTM1, ETV6*	DNA and RNA NGS & IHC
Ou, 2019 [56]	NSCLC	0.1%	*NTRK1-3*	*IRF2BP2, TPM3,* and others	DNA NGS
Xia, 2019 [57]	NSCLC	0.056%	*NTRK1*	*CD74, IRF2BP2, LMNA, PHF20, SQSTM1, TPM3, TRP*	DNA NGS

NTRK, neurotrophic receptor tyrosine kinase; RT-PCR, reverse transcriptase polymerase chain reaction; NGS, next generation sequencing.

**Table 6 biomedicines-10-02531-t006:** Neurotrophic receptor tyrosine kinase (NTRK) gene fusion in lung cancer detected using NGS at DNA and RNA levels.

Histology	DNA Rearrangement	RNA Fusion
**ADK**	*C14orf2*-*NTRK1* (intergenic: intron 11)	*KIF5B*-*RET* (exon 15–exon 12)
	*RRNAD1-NTRK1* (UTR3-exon 15)	*TPM3*-*NTRK1*(exon 8–exon 10)
	*NTRK1*-*NBPF25P* (intron 8-intergenic)	*TPM3*-*NTRK1*(exon 8–exon 10)
	*NTRK1*-*ARHGEF11* (exon 17: intron 1)	*SQSTM1-NTRK2* (exon 4–exon 15)
	*NTRK1*-*FMN2* (intron 11: intron 16)	*KIF5B-NTRK2* (exon 24–exon 15)
	*TPM3*-*NTRK1* (intron 8: exon 9)	
	*TPM3*-*NTRK1* (intron 8: intron 9)	

## Data Availability

Not applicable.

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
