# Peer review of "Adenocarcinomas of the Lung and Neurotrophin System: A Review"

_biomedicines, 2022, doi:10.3390/biomedicines10102531_

Round 1

Reviewer 1 Report

The article is informative and important in clinical practice. My suggestion is to present a table on which the authors present the list of commercial tests (genetic fusion probes or antibodies) to detect the presence or activity of neurotrohins in lung adenocarcinoma.

Minor spell check and punctuation is required.

Author Response

I thank the reviewer for the helpful comments that allowed us to implement the quality of the manuscript:

  1. My suggestion is to present a table on which the authors present the list of commercial tests (genetic fusion probes or antibodies) to detect the presence or activity of neurotrohins in lung adenocarcinoma

We have added in the text different tables to lists some commercial products used for the detection of NTs and NT receptors at protein and RNA levels: primers (Table 2), antibodies (Table 3) and the most frequent gene fusion detected by NGS and RT-PCR for DNA and RNA (Table 6)-.

A grammar revision has been done

Reviewer 2 Report

The review is comprehensive and relevant. The review article includes a clear and concise abstract. The introduction sets the scene by describing all the recent findings to uncover the perspective of adenocarcinomas of the lung and neurotrophin system. However, the key message is not conveyed in the illustrations. Additionally, some more information about the overexpression of NGF and the increased release of hypoxia-inducible factor-1alpha in context of  the tumor neo-angiogenesis should be added. The methodology for literature search is not described. The spell check and grammar should be performed. 

Author Response

I thank the reviewer for the helpful comments that allowed us to implement the quality of the manuscript:

  1. Some more information about the overexpression of NGF and the increased release of hypoxia-inducible factor-1alpha in context of  the tumor neo-angiogenesis should be added.

The role of hypoxia-inducible factor-1 alpha as and its involvement in neo-angiogenesis as well as in NT system activation was added and emphasized (line 72)

  1. The methodology for literature search is not described.

The methodology for the literature search was added in the text and in a specific table (Table 1) by which this strategy was conducted.

  1. The spell check and grammar should be performed

Grammar and spell check revision have been done

Reviewer 3 Report

Dear authors, the review posess merits. However, I consider that some points still should be better explored.

The methodology used is not clear. What are the bibliographic bases used? What are the inclusion criteria? And you of exclusion? What or number of items used in the review?

Please refer to the review proposal below:

"The purpose of this review was to extensively analyze the literature data in order to provide a clear view of the available data in order to clearly cover novel molecular mechanisms useful for diagnostic but, above all, therapeutic purposes."

The authors will highlight the aspects or molecular markers, but in a disorganized way, not referring to the types of non-lung cancers. I suggest following an order of mentioning each marker and then mentioning "everything" about each type of lung cancer at once. Assim, apart from the therapeutic aspect is not clear. Apart from pharmacology (dose, toxicity, use in humans) the drugs that can be used are scarce or non-existent. This contradicts the purpose of the review.

The pharmacological intervention against NT signaling may be essential not only to directly counteract cancer cell biology, but also indirectly modulate in an inhibitory way, neurogenesis and/or angiogenesis with potential impact on tumor growth and progression.

Some information about this point is poorly mentioned.

I suggest not topically highlighting each molecular object, each drug that can be used therapeutically. Additionally, the authors can indicate where this drug is, what aspects of cancer progress it controls, highlight whether the drug has been used in humans for other purposes or not, wonder about its toxicity, at the dose used. With these elements gathered, a real reflection on the therapeutic use can be established.

Author Response

I thank the reviewer for his criticisms, which I find extremely useful, improving the manuscript with providing data that clarify NTs clinical significance and underlining possible therapeutic applications. This has allowed us to highlight the research that has resulted, from the initial studies of NTs anatomical localization within neoplastic lung samples to the following pharmacological applications. This approach is essential and the discovery of novel molecular target allow us to develop additional biological therapies.

  1. The review proposal was better clarified as the reviewer suggested.

Your suggested phrase to better define the review proposal was inserted in the text lines 94-101. Furthermore, methodology used was added in a table and in the text (Table 1) to clarify our research strategy

  1. The authors will highlight the aspects or molecular markers, but in a disorganized way, not referring to the types of non-lung cancers. I suggest following an order of mentioning each marker and then mentioning "everything" about each type of lung cancer at once. Assim, apart from the therapeutic aspect is not clear. Apart from pharmacology (dose, toxicity, use in humans) the drugs that can be used are scarce or non-existent. This contradicts the purpose of the review.

There is now a paragraph about NTs and cancer (paragraph 2) showing an overview on the role of these proteins in different cancer types.

Moreover, we have better systematized the chapters on NTs in ADK distinguishing the pan-receptor p75 the receptor able to bind all types of NTs from each system consisting of NTs and their specific receptors, NGF and TrkA; BDNF / NT4 and TrkB, NT3 and TrkC. In this way it might be easier to consult.

  1. Therapeutic aspect is not clear. Apart from pharmacology (dose, toxicity, use in humans) the drugs that can be used are scarce or non-existent.

We wrote a new section about treatments (Section 4) including dosages, side effects and indications about approved ones (Entrectinib and Larotrectinib). In conclusion, Table 4 summarizes  the ongoing clinical trials in NTRK fusion positive solid tumors including NSCLC.

Round 2

Reviewer 3 Report

Dear authos, I suggest that the authors add the information on how the methodology was created.

What criteria are used to select the manuscripts?

What are the criteria to discard the manuscripts?

How many foram manuscripts used in the review?

I suggest inserting the clinical information in the Abstract.

Author Response

Dear Reviewer

Please find the answer to your questions below.

We thank you very much for your suggestions that allow the manuscript to improve. 

1. The criteria used

Inclusion criteria used to select manuscripts derived from the PICO (problem, intervention, comparison and outcome) and PIPOH  (population,  interven-tion(s),  professionals,  outcomes,  and  health  care  setting/context) structures. Therefore, studies used are all experimental that have been published in the last 20 years.  

2. Criteria to discarde manuscripts:

Exclusion criteria are the studies published in a language other than English and the use of a qualitative methodology.

3.How many manuscripts are used in the review

About 190 manuscripts are selected and evaluated for the review. The papers used are listed in the reference section.

4. Clinical information in the abstract

Several clinical trials on humans are ongoing using NT receptor antagonists in different cancer cell types for further therapeutic applications.  (in the abstract).